# Qualitative Study to Explore the Occupational and Reproductive Health Challenges among Women Tobacco Farm Laborers in Mysore District, India

**DOI:** 10.3390/ijerph21050606

**Published:** 2024-05-09

**Authors:** Priyanka Ravi, Kiranmayee Muralidhar, Maiya G. Block Ngaybe, Shivamma Nanjaiah, Poornima Jayakrishna, Ashley A. Lowe, Karl Krupp, Amanda M. Wilson, Frank A. von Hippel, Zhao Chen, Lynn B. Gerald, Purnima Madhivanan

**Affiliations:** 1Department of Health Promotion Sciences, Mel & Enid Zuckerman College of Public Health, University of Arizona, Tucson, AZ 85721, USApmadhivanan@arizona.edu (P.M.); 2Public Health Research Institute of India (PHRII), Mysore 570020, Karnataka, India; 3Advanced Nursing Practice and Science Division, College of Nursing, University of Arizona, Tucson, AZ 85721, USA; 4Asthma & Airway Disease Research Center, University of Arizona Health Sciences, Tucson, AZ 85724, USA; 5Public Health Practice, Policy, and Translational Research Department, Mel & Enid Zuckerman College of Public Health, University of Arizona, Tucson, AZ 85721, USA; 6Community, Environment & Policy Department, Mel & Enid Zuckerman College of Public Health, University of Arizona, Tucson, AZ 85721, USA; amwilson2@arizona.edu (A.M.W.);; 7Epidemiology and Biostatistics Department, Mel & Enid Zuckerman College of Public Health, University of Arizona, Tucson, AZ 85721, USA; 8Office of Population Health Sciences, Office of the Vice Chancellor for Health Affairs, University of Illinois, Chicago, IL 60612, USA

**Keywords:** focus group discussion, occupational health, reproductive health, tobacco farmers, tobacco farming

## Abstract

Tobacco farm laborers are primarily women and children working for very low wages. The aim of this study was to explore occupational and reproductive health challenges faced by women tobacco farm laborers in Mysore District, India. We conducted interviews and six focus group discussions among 41 women tobacco farm laborers. Codes and themes were generated based on deductive and inductive approaches using the socioecological model. Participants reported symptoms of green tobacco sickness including headaches, back pain, gastric problems, weakness, and allergies during menstruation, pre-natal, and post-natal periods. Participants had poor awareness about the health effects of tobacco farming, and there were gender inequalities in wages and the use of personal protective equipment. Participants received support from family and community health workers during their pregnancy and post-natal period. Women reported wanting maternity benefits from the tobacco board, as well as monetary support and nutritional supplements. There is a need for health education about the environmental dangers of tobacco among farm laborers, and more supportive policies for women farmworkers during pregnancy and post-natal periods.

## 1. Introduction

India is the second largest tobacco growing nation globally, producing over 772,152 tons of leaves in 2022 [1]. The World Health Organization (WHO) Framework Convention of Tobacco Control (FCTC) article 17 recommends promoting economically viable alternatives for tobacco workers and growers, and article 18 recommends protection of worker health and the environment with respect to tobacco cultivation [2].

Green tobacco sickness (GTS), an occupational illness associated with tobacco, is commonly seen among workers engaged in tobacco cultivation [3]. The illness was first reported in 1970 among tobacco workers in Florida, as “cropper sickness” [4]. Later, the symptoms were found to be caused by the absorption of nicotine from wet tobacco plants and reported as GTS [4]. The use of personal protective equipment (PPE) such as water-resistant clothing, chemical-resistant gloves, plastic aprons, and rain-suits with boots is a recommended preventive measure that should be used by tobacco farmers to prevent GTS [3]. Symptoms of GTS include chills, headaches, nausea, vomiting, pallor, dizziness, increased perspiration, diarrhea, abdominal pain, and increased salivation, among other shorter term symptoms based on exposure [3]. Morbidity and mortality rates of GTS around the world have not been widely documented; a morbidity estimate conducted in 1994 based on reports to a poison control center in Kentucky found that approximately one in four people with suspected GTS were admitted to the hospital [5]. Another cross-sectional study in Brazil found 122 (34.5%) cases of GTS among tobacco workers included in their study, 39% of whom were smokers and 61% male [6]. A review found that the prevalence of GTS varies globally from 8.2 to 47% [3]. GTS symptoms were reported among children of 12 to 17 years of age in Kentucky Regional Poison Center [7]. A case–control study conducted in 1993 reported children younger than 16 years of age represent 9% GTS reported [8].

Prevalence of GTS by gender has varied in the literature, some finding it higher among men and others higher among women [4,5]. However, the prevalence of GTS was found to be higher among women tobacco farmers compared to men in India and Korea [9,10]. It is also possible that the burden among women is underrepresented at times due to the historical lack of female workers in tobacco [11]. During tobacco production, women are affected differently from men biologically due to the critical phases of vulnerabilities during pregnancy [12]. In Brazil, among women tobacco farm workers with a family history of asthma, wheezing was associated with tying bundles of tobacco, strenuous work, contact with chemical disinfectants, and GTS [13]. Women additionally have a higher burden of stressors from of tobacco work [14]. Among women tobacco farmers in Ambulu village in Indonesia, a significant relationship was observed between work stress and menstrual disorders [15]. Almost 5% of infants whose mothers exposed to pesticides in tobacco farming failed hearing screening. [16]. Tobacco farmers exposed to pesticides also exhibited signs of central auditory dysfunction characterized by decrements in temporal processing and binaural integration processes [17]. 

Pesticides are essential for tobacco growing, as “the crop could not be produced economically without them” according to tobacco industry documents [18]. Farmers are exposed to pesticide residues through ingestion, inhalation, and skin contact with dry tobacco leaves during work [19]. Organochlorine and organophosphate pesticides are widely used in tobacco farming [20]. Dichlorodiphenyltrichloroethane (DDT) is an organochlorine pesticide known for its long-term persistence in the environment and its irreversible toxic health effects causing neurological damage, endocrine disorders, and reproductive health effects [20,21]. Among tobacco farmers using pesticides, 12.2% received a medical diagnosis of poisoning [22]. Acute pesticide poisoning is associated with the number of exposure types, multi-chemical exposure, clothes being wet with pesticides, and spillage on the body/clothes [22]. 

Occupational exposure to pesticides among tobacco workers increases the frequency of adverse pregnancy outcomes such as gestational hypertension, fetal growth restriction, premature birth, and low birth weight [23]. Some pesticides are endocrine-disrupting chemicals (EDCs), which mimic hormones or otherwise disrupt hormone axes [24]. Many EDCs are persistent in the environment and bio-accumulate [25]. Some EDCs mimic the female hormone estrogen which can result in hormone-related cancers such as breast cancers [26]. Exposure to DDT in utero is also associated with an increased risk of breast cancer [27]. Tobacco farmers may experience compromised deoxyribonucleic acid (DNA) integrity associated with enhanced oxidative stress levels during the harvest and grading period [28]. Pesticide-exposed workers have more DNA damage revealed by the comet assay and micronucleus test than do non-exposed workers [29]. Deficiencies of folate, vitamin B12, and vitamin B6 were associated with genotoxic effects in tobacco farmers exposed to pesticides [30]. However, despite these vulnerabilities specific to women working with tobacco, the literature on the health of women tobacco farmers in India is sparse. 

In Southern India, there has been some evidence of the occupational impacts of tobacco cultivation. For example, a study in Andhra Pradesh, India, found evidence of occupationally derived health issues among tobacco farmworkers including body aches, dermatitis, and back pain from physical hazards; toxicity, infection, rashes, skin lesions, and respiratory problems from chemical hazards; and the strain and irritation of eyes from biological hazards [31]. Another study in Tamil Nadu, India, found that exposure to pesticides among farmers may increase risk of issues such as excessive sweating, burning/stinging/itching eyes, and runny/burning nose [32]. Another study in Hassan District, Karnataka, further compared tobacco farmers to non-tobacco farmers, finding that tobacco farmers experience significantly more negative symptoms associated with GTS such as nausea, dizziness, increased salivation, poor appetite, insomnia, and increased sweating [33]. Another similar study in south-eastern Bangladesh reported similar findings both among adults and children [34]. Information is lacking on the impacts of tobacco farming in Southern India on sexual and reproductive health.

India has some protective mechanisms to support tobacco workers, such as manuals and trainings to encourage the use of personal protective equipment among workers by institutes such as ICAR (Central Tobacco Research Institute) and the Indian Ministry of Labour and Employment [35,36]. However, it has been recognized that in practice, many may not actually use their personal protective equipment, even if available, perhaps due to a lack of understanding of the health impacts of tobacco and pesticide handling [33,34].

In addition to health effects for farm workers, tobacco production results in many other impacts on local communities and the environment. Tobacco cultivation results in detrimental effects on surrounding ecosystems, including deforestation resulting from the demand for wood to cure tobacco leaves, degradation of soil fertility, pollution of ground and surface water, and adverse impacts on biodiversity due to the intensive use of chemical fertilizers and pesticides [37,38]. From a socioeconomic perspective, farmers often have contractual arrangements with the tobacco industry and are trapped in debt [39]. Additionally, in some countries, children from poor households miss school to work in tobacco farming [40]. Furthermore, tobacco dependence among tobacco farmers is often high compared to non-tobacco farmers [33]. 

The aim of this study is to qualitatively explore the challenges in occupational and reproductive health of women tobacco farm laborers across their lifespan. The objectives are, among non-pregnant women working in tobacco farms, to (1) determine reproductive health support and challenges using the socioecological model (SEM) across the life course; (2) determine occupational health support and challenges; and (3) understand the knowledge, attitudes, and practices towards safe work environments.

## 2. Materials and Methods

### 2.1. Socioecological Model

This is an exploratory study using interviewer-administered questionnaires and focus group discussions (FGDs) guided by the socioecological model (SEM) to explore the beliefs, attitudes, and practices of women tobacco farm laborers with regard to the social, environmental, and health risks of tobacco farming. The SEM is a framework for explaining the sphere of influence that affects human behaviors. The model dictates that health-seeking behavior is a product of interactions among individual attributes and environmental factors. According to SEM, five spheres of influence affect human behavior: intrapersonal, interpersonal, community, organization, and public policies [41]. 

Intrapersonal factors include characteristics of the individual such as knowledge, attitudes, behavior, self-concept, skills, etc. This includes the developmental history of the individual. Interpersonal processes include formal and informal social networks and social support systems, including the family, work group, and friendship networks. Institutional factors are social institutions with organizational characteristics and formal (and informal) rules and regulations for operation. Community factors are relationships among organizations, institutions, and informal networks within defined boundaries. Public policy includes local, state, and national laws and policies [41].

### 2.2. Study Setting

This study was conducted from May to July 2022 in six randomly selected villages in Hunsur *Taluk*, Mysore District, India, a major tobacco growing area in the southern state of Karnataka. This study was conducted at the *Anganwadi* centers (rural childcare centers) and the community centers where residents usually gather. The FGDs were conducted outside of regular working hours at a time convenient for the participants.

### 2.3. Participants and Recruitment

The choice of women farm laborers as study subjects was due to their role in tobacco production, combined with their essential role in providing care for the family, especially related to the health of children and the elderly. A representative from the Public Health Research Institute of India (PHRII) first consulted with community members and reached out to community health workers known as Accredited Social Health Activists (ASHAs) in the communities where the planned recruitment of participants would happen. The ASHA workers living in these villages are well connected to the residents and facilitated the recruitment of potential study participants. A flyer about the study was distributed by the ASHA workers during their regular house visits. PHRII has previously conducted several studies in these communities and developed good communication with these communities. ASHA workers in each village were provided monetary incentives of INR 300 (3.75 USD) for their time in community engagement.

Inclusion criteria for participation in the study were women tobacco farm laborers in Mysore District, who were 18 years and older and had experienced pregnancy in the past irrespective of the birth outcome. We included participants who understood and spoke Kannada, consented to audio recording, belonged to different households, and had the ability to undergo the informed consent process. Exclusion criteria were currently pregnant women as they are a vulnerable group, women who have never been pregnant, and those not willing to provide written consent and consent for audio recording.

This study was approved and monitored by the Institutional Ethics Review Board at PHRII (IERB Protocol number #2022-05-07-68, 7 May 2022). The University of Arizona IRB confirmed reliance for this study on the external IRB at PHRII as the IRB of Record (STUDY00001243, 8 June 2022).

### 2.4. Study Sampling

The participants were recruited through convenience sampling with the help of the ASHA workers and the staff from PHRII. Participants in different age groups were included to capture the different reproductive health concerns across age groups. Six FGDs were conducted with a total of 41 participants (6–7 individuals per group). The consent form was read aloud to the participants in the local language of *Kannada*, and written informed consent was obtained before the start of the study. 

This study consisted of an interviewer-administered questionnaire and six FGDs among 41 participants. Those who agreed to participate were asked questions about their occupation, reproductive health, and tobacco farming experience and challenges. 

### 2.5. Approach and Recruitment

The interview was conducted on the first day of the study to develop rapport and trust among the participants. The interview took 15 min per participant. All participants were asked questions regarding demographic characteristics, marital status, religion, caste, socioeconomic status, self-reported smoking and use of smokeless tobacco and areca nut, exposure to second-hand smoke, tobacco farming during pregnancy including exposure to chemicals and pesticides, allergies, allergies of children, and use of personal protection equipment (PPE) at work (Appendix A). 

### 2.6. FGD Data Collection

FGDs were conducted to explore occupational and reproductive health issues. The FGDs ranged from 60 to 70 min each. The FGD guide (Appendix A) was based on the domains of the SEM [41]. The FGD guide was pilot tested twice among PHRII staff before use in the field. On the day after the recruitment, an interviewer-administered questionnaire was answered by a group of participants, and the FGD was administered among the participants. The participants were asked about occupational and reproductive health issues and experiences at the individual, interpersonal, community, organizational, and societal levels. The FGD questions assessed access to health care services during pregnancy; how working in tobacco farming would have affected reproductive health with a focus on pregnancy, menstruation, and menopause; exposure to pesticides and chemicals during pregnancy; and access to PPE. The questions also explored support from family, friends, doctors, other health care workers, the farming community, and local, state, and national governments. The facilitator and notetaker were women to help participants feel more comfortable answering questions. The facilitator was fluent in *Kannada* and used the FGD to initiate the discussions, and probing questions were asked to explore more on a particular topics. A notetaker took notes and observed the participants’ non-verbal communications. The FGDs were audio recorded and transcribed verbatim to *Kannada*. Data collection continued until data saturation was reached [42]. A trained translator later translated the transcripts from *Kannada* to English for analysis. We did not do back translation due to limited resources. All participant identifierswere removed, and only de-identified data were used for coding and analysis. Participants were paid a stipend of INR 250 (3.12 USD) in cash for their participation, and refreshments were served at the end of the FGDs.

### 2.7. Trustworthiness

Peer debriefing was conducted by the field researchers PR, SN, and MBN after every FGD with non-field coresearchers KM and PM with the notes taken during the discussions. These discussions helped us to identify the repeated themes and decide on the data saturation. During the study period, the field researchers visited the tobacco farms and tobacco barns to observe the tobacco farming and tobacco processing activities. Data triangulation in FGDs was performed with five groups of tobacco farm laborers not owning tobacco fields and one group with tobacco farm owners who also work in the tobacco fields. The audio recordings, notes, and transcripts are available for audits at PHRII. 

### 2.8. Research Team and Reflexivity Statement

Researchers based at PHRII have intricate knowledge of the local community, language and culture; therefore, the author SN from PHRII facilitated all of the FGDs. PR was the study’s principal investigator and notetaker in all the FGDs. Authors PR, MBN, and KM, who conducted the qualitative analysis, are researchers at the University of Arizona and PHRII, and all hold graduate degrees in an area of health. MBN was a fellow PhD student with previous experience in qualitative research, and KM was a research physician at PRHII. All researchers conducting data collection and data analysis were female. Authors AAL, KK, AMW, LBG, ZC, and PM have PhDs in public health fields of epidemiology, health promotion, and environmental health sciences. A relationship was established prior to study commencement with the study participants through PHRII staff, who have an extensive history of research among these populations. Participants were familiar with the facilitator. The facilitator and notetaker did not provide their opinions during the FGDs verbally or non-verbally.

### 2.9. Data Analysis

We used MAXQDA 22 software, VERBI Software, Berlin, Germany for qualitative data analyses [43]. We used a thematic content analysis approach. Two investigators (PR and MBN) independently developed coding trees based on the SEM using the first interview, which were then consolidated into a codebook through discussion and entered into MAXQDA. The codebook was iteratively reviewed throughout the rest of the coding process. They then coded the data and compared for discrepancies through discussion to improve interrater reliability. A third investigator (KM) helped to resolve any discrepancies which could not be resolved through discussion. Codes and themes were generated based on an inductive approach using content analysis and a deductive approach using the SEM model. We used consolidated criteria for reporting qualitative research (COREQ), including a 32-item checklist for focus groups as a standard qualitative study reporting guideline [44].

## 3. Results

### 3.1. Participant Characteristics

The total sample consisted of 41 participants. The mean age was 38.22 ± 7.95 years, and more than half of the participants were 31 to 40 years of age (22/41, 53.7%). The majority of participants were married (33/41, 80%), belonged to the Hindu religion (41/41, 100%), and self-identified as “scheduled caste” (30/41, 73%). The caste system in India is a complex social structure, in which the scheduled caste are officially regarded as socially disadvantaged and occupy the lowest step of the social ladder [45]. Almost half of the participants were illiterate and lacked a formal education (19/41, 46%), and 61% of participants had an annual income of (240 to 600 USD) INR 20,000 to 50,000 (25/41). None of the participants reported using any smoking form of tobacco, but 15% reported using smokeless tobacco (6/41) (Table 1).

### 3.2. Occupational and Reproductive Health

The second part of the interview focused on the reproductive and occupational health of the participants. Most participants were involved in tobacco farming activities like cultivation (36/41, 87.8%), harvesting (40/41, 97.6%), sorting (36/41, 87.8%), stacking (33/41, 80.5%), and packing (22/41, 53.6%). One participant reported having had multiple abortions (1/41, 2.4%), and five participants reported having had a miscarriage (5/41, 12.2%). Four participants reported death of the infant before reaching one year of age (4/41, 9.8%). Two participants reported using tobacco during pregnancy (2/41, 4.9%). Four-fifths of participants were involved in tobacco farming during pregnancy (33/41, 80.5%). Eighteen participants reported being exposed to tobacco dust during pregnancy (18/41, 43.9%). Only six participants reported wearing PPE during work with tobacco (6/41, 14.6%). The history of any allergy was reported in less than one quarter of participants (Figure 1; Table 2).

### 3.3. Qualitative Analysis

Throughout the results, data extracts are presented with the participant number (e.g., P1) and FGD number (e.g., G1). Most participants were involved in all the tobacco farming activities including sowing, harvesting, sorting, and packing. For example, one participant said, 


*“We tie the tobacco leaves, cut the leaves, manure the leaves, dig holes for the saplings and do the bridging.”*
(G4, P1)

Another said, 


*“I do the same work as others, planting the saplings, putting in the fertilizers, removing the weeds, pushing the mud, tying the leaves and the rest of the work.”*
(G5, P4)

Figure 2 represents the major themes that reflect occupational and reproductive health experiences for women tobacco farmers across the five levels of the SEM.

#### 3.3.1. Intrapersonal

##### Health Beliefs

The tobacco plant was believed to be the Hindu deity of wealth ‘Lord Lakshmi’, which impacted the behavior of workers. One participant noted,


*“Tobacco is considered as goddess Lakshmi so during the menstrual periods we do not touch the tobacco plants until we take a shower.”*
(G4, P4)

Participants believed that working in the tobacco fields and touching tobacco leaves while grading their quality produces heat, sucks the blood, and makes them look pale and anemic. One participant mentioned, 


*“It is heat and it also sucks blood.”*
(G4, P4)

Some participants believed tobacco use can cause cancer; however, not all participants believed working in tobacco will cause any harm. Some participants reported using tobacco with betel quid for relief from toothaches and when they are bored. For example, several participants stated, 


*“They eat it when they get toothaches, it reduces the pain.”*
(G3, P4)


*“Chewing tobacco is a habit; people eat it because they get bored.”*
(G3, P6)


*“Now so many people are facing many problems by having tobacco, by smoking the beedis so many people have got kidney problems, doctors tell them to stop smoking, but they say yes there, and will not follow it in the house, but then they continue to smoke.”*
(G1, P3)

##### Health Behaviors

Hand washing after work and using PPE were some of the health behavior practices discussed in the FGDs. Most participants reported washing hands after work with soap, detergents, and tamarind water. However, one participant reported not washing hands because washing hands took time: 


*“I used to feed my child after work, without washing my hands because washing it would take a lot of time.”*
(G2, P3)

Overall, participants were not interested in using PPE because their use reduced working efficiency. For example, one participant said, 


*“We cannot pluck the leaves if we wear all that. It will be difficult to work for us when we wear gloves. It keeps on sliding down from our hands and we cannot pluck the leaves quickly and we will feel like falling if we wear the boots.”*
(G2, P3)

However, in one focus group, participants reported wearing long-sleeved shirts and tying a cloth to cover their head to avoid direct exposure to sunlight. One participant noted, 


*“We wear a long shirt that covers our hands and also tie a cloth to cover our heads, we work after doing this.”*
(G2, P2)

One focus group participant reported that after the COVID-19 pandemic, the tobacco board provided them with gloves, masks, and boots and took their picture wearing these, but that they never subsequently used this PPE. 

Most focus groups reported that PPE was provided to the tobacco farm owners and, sometimes, to the men spraying pesticides in the field. Participants reported male tobacco farm laborers had greater access to PPE. Women tobacco farm laborers generally did not understand the purpose of PPE and thought that its use was restricted to pesticide sprayers and used for the COVID-19 pandemic. 

##### General and Occupational Health Experiences

Participants reported recurring and chronic GTS symptoms such as nausea, vomiting, headache, stomach pain, musculoskeletal pain, back pain, gastric problems, weakness, and allergies during work. They reported that these problems were temporarily resolved while taking medicines. Participants experienced GTS symptoms during menstruation and during pre-natal and post-natal time periods. To seek relief from the pain, in most cases, participants self-medicated by using over-the-counter drugs and home remedies such as tea. If symptoms were not relieved, then, they consulted doctors.

One participant noted, 


*“I usually get headaches three to four days a week for the past three years. I used to apply headache balm, took medicines from the medical stores, or took injections from the hospital. I cannot do any work if I get headache. I will usually get it during evening hours, and during the morning. The last few days if I get it in the morning it stays till evening. The pain reduces only after getting an injection from the hospital.”*
(G5, P4)

Another participant said, 


*“Sometimes I go to the hospital because of body ache. My blood pressure will be low. Doctors tell me not to expose to the sun for long, not to work more and take rest. They tell me that my health is good, and I should take tablets for low blood pressure. In the last 2-3 years after my child was born, they told me that I have less blood (anemia). If they give blood and a tonic, I will be fine. They give tonic and powder (supplements) in the government hospital. I take it. I have a body ache and then the regular common cold. Other than that, I do not have any disease. This is common, I get itching, that is all.”*
(G4, P3)

Another participant said,


*“We have to do the tobacco work. It is compulsory and we cannot sit simply without doing it. We must go together with our husbands. I will be questioned if I am late. Sometimes when I get a headache I will be having no time to rest even for five minutes. I will take the tablet but cannot rest, and they will be screaming and will not understand our pain. They will ask us to pack the boxes and will scold us if we will be late, even though we do not take the medicine properly. Also, we must be on time with them.”*
(G1, P5)

##### Reproductive Health

The reproductive health experiences of woman tobacco farm laborers during their menstruation, pregnancy and childbirth, post-natal period, and menopause, as well as other reproductive concerns, were discussed.

Menstruation

Most participants reported having regular menstrual periods ranging from three to five days a month. One participant shared, 


*“I have a regular menses. Sometimes it will be for three days and sometimes for five days. I have no problems except lower back pain.”*
(G3, P4)

A few participants reported irregular periods, either unusually infrequent or unusually frequent. For example, one participant said, 


*“I get my menses once in six or seven months. Even during that time, there will not be heavy bleeding.”*
(G2, P3)

Another participant said, 


*“For the past one month I am having more bleeding, in the same month I got my periods three times. I am almost changing ten pads a day and it happens for six days. This month I had it four times. I had very severe bleeding this month and due to that my eyes became blurry, and I also feel giddiness. I feel very tired all the time and I feel uneasy to lift my head.”*
(G2, P5)

Participants used cloth and sanitary pads as menstrual hygiene products. One participant responded, 


*“We tie a cloth to cover our head while working. We wear that cloth as a pad if we get periods at work. We cannot speak loudly because there will be five other people. We do not disclose it to all. We adjust among ourselves. When someone get their periods, we make them sit separately and make them work. We will not make them move around much.”*
(G4, P1)

Participants reported working in the tobacco fields and doing domestic work during menstrual periods. Symptoms of dysmenorrhea and GTS were reported during menstruation. As one participant noted, 


*“We used to do the same tobacco work even during that time. I used to get stomach pain and lower back pain after I came from work. Despite that, I do the household work and cooking at home.”*
(G2, P5)

Some farm owners do not allow women with menstrual periods to work. For example, one participant said, 


*“Some will tell us, nothing will happen, you can come. Some tell us not to come. They will not allow us to touch. Some will not say anything.”*
(G4, P4)

Pregnancy and childbirth

Almost all the participants were involved in tobacco farming activities during their pregnancy. Most participants worked until six to eight months of pregnancy, and a few worked until the day of delivery. For example, one participant stated, 


*“I did the tobacco work until three days to my delivery. I got my cramps but still I was working.”*
(G4, P3)

They reported experiencing GTS symptoms such as weakness, nausea, vomiting, and headache during pregnancy. However, it is unclear if the vomiting and nausea were pregnancy related or GTS symptoms. Participants also mentioned being exposed to fertilizers during pregnancy. Some participants reported adverse birth outcomes including miscarriages, stillbirths, and low birthweight (LBW) infants. One participant noted, 


*“During pregnancy, I used to feel like vomiting while doing the tobacco work. I have vomited while putting the fertilizers.”*
(G2, P4)

Another participant said, 


*“We used to bend and do the activities like planting the saplings until we came to our 5th month. After the 5th month, the stomach starts to increase, so we sit and do works like tying the leaves and separating them.”*
(G2, P3)

Post-natal period

Most participants returned to work three months to one year post-delivery. Participants spoke about their experiences during the post-natal period with breast feeding the infant and not having a family member for childcare service. Few participants used *Anganwadi* centers for childcare service. One participant said, 


*“If there was someone to take care of the baby, we would leave the child and go for work; otherwise, we would carry the baby. Aged people will take care. Either father-mother or father-in-law and mother-in-law. Now, we must take care of our children. If there is work, we have made the baby sleep there and do our work.”*
(G4, P4)

Another participant noted, 


*“After my first delivery I started my work in three months. I worked keeping the baby next to me. Sometimes, I would leave the child in the anganwadi (childcare center) from morning to evening. I have suffered a lot.”*
(G4, P1)

Other reproductive health experiences

Participants experienced other reproductive health problems such as excess vaginal discharge, itching, and redness. For example, one participant said, 


*“White discharge started during my last childbirth. Now my son is 16 years old, and I am having it since then. Initially I had little white discharge and along with that I had itching as well. I showed it to the doctor, and they said it is very common in women and they said there is no problem in it. There was smell there in the beginning. Now the itching has reduced a little bit after being consulted with the doctor 2-3 times. But it reoccurs after some time, and it reduces again after washing in hot water.”*
(G2, P6)

Menopause

Four participants had gone through menopause. Three of them had natural menopause, and one participant had undergone a hysterectomy. Participants reported having longer menstrual periods (up to one month) during the beginning of menopause; however, now they do not have any problems while working. For example, one participant said, 


*“I had lower back and stomach pain. I could not sit down, and I used to find it very difficult. I had been bleeding for five days. I had lot of stomach pain and bleeding for a month on my last period. They gave me an injection saying that it would be like this at the beginning of menopause. Since 1 ½ years, I have not got my period. I am working the same as earlier. I do all the household work at home without a single day leave. I even work at the fields and even go to daily wage work.”*
(G5, P7)

##### Substance Use

A few participants reported using chewing tobacco. Two of the participants used tobacco during pregnancy and believed using chewing tobacco during pregnancy does not cause any harm. For example, one participant said, 


*“I eat chewing tobacco (kaddipuddi) along with betel nut and leaves two to three times a day. I keep it in one corner of the mouth and will spit it after some time.”*
(G2, P1)

Another participant said, 


*“I have been chewing tobacco for the past 25 years and nothing has happened to me. I eat 15 times per day. I don’t know why I started, but I used to get teeth pain and swelling. At that time the old people in the village asked me to use this for reducing pain, and then I started having it.”*
(G5, P4)

#### 3.3.2. Interpersonal

##### Family Support

Some participants received support from husbands, parents, siblings, and in-laws during their pregnancy and the post-natal period. For example, a participant said, 


*“My husband helped me during pregnancy and delivery. He did not send me to my father’s house. My husband took care of me.”*
(G4, P2)

Another participant said,


*“During that time, I used to be in my mother’s place; my sisters-in-law and my mother used to be there for my support.”*
(G5, P3)

A few participants did not receive support from their families during pregnancy. For example, one participant said, 


*“No one helped us during that time. We were doing all the work. People would feel jealous. I would bring water even nine months pregnant, do the cooking, cleaning, washing vessels. Everybody would go out to work.”*
(G4, P1)

##### Friends and Neighbors

Support from friends and neighbors was sparse. When the family was unavailable, one participant reported receiving support from the neighbors for hospital visits; she stated, 


*“When I was pregnant, I had fallen down and my neighbor helped me and took me to the hospital.”*
(G1, P1)

None of the participants mentioned receiving support from their peers or friends.

##### Family Health

Most of the family members were also involved in tobacco farming activities. Some of the most common health problems reported among the participant’s family members include hypertension, thyroid problems, gastric problems, and respiratory problems such as asthma. For example, one participant said, 


*“My husband has pain in one of his thighs and blood pressure, for the last four to five years it is increasing. He also works in tobacco fields”*
(G2, P7)

Another participant said,


*“My mother-in-law is not well. She has breathing problems like asthma. She was doing tobacco work. Now she cannot stand the smell of tobacco.”*
(G4, P3)

One participant reported acute symptoms while spraying pesticides.

##### Substance Use in Family

Alcohol, smoking beedi, and chewing tobacco products like *khaini* and *ghutka* are some of the common substances used by the participants’ spouses. For example, one participant mentioned, 


*“My husband will drink alcohol and smoke beedi.”*
(G2, P6)

Participants would benefit from addiction counseling services in the community.

#### 3.3.3. Organizational

##### Wages

The wages of tobacco farm laborers range from INR 200 to 300 per day for women and INR 300 to 500 per day for men, depending on the type of work. This gender inequity in wages exists despite the equal number of hours worked by women and men. Some landowners provide men with alcohol in addition to their salary, which causes alcohol abuse and addiction problems in some households. As mentioned by a participant, 


*“Men are paid Rs. 300 and women are paid Rs. 200. Along with Rs. 300, men will be given alcohol. If we ask them for more wages, they will start fighting with us. The prices of daily needs have been raised. One litre of oil costs Rs. 200 and they tell us that they do not have money growing in plants to give us wages whenever we ask. The women will be satisfied even if they are paid Rs. 100. They will just take the money and will not question about it.”*
(G3, P4)

##### Tobacco Board

At the organizational level, participants did not receive any maternity benefits from the farming organizations and tobacco board. A participant stated, 


*“Support must be given from the tobacco sector, as we are working for them for many years. If they provide such support, it will be even more helpful and we will also be happy. If they conduct health camp from the tobacco sector, it will help us.”*
(G1, P7)

Another participant said,


*“We have not got any facilities madam. Those who built the tobacco barn have got some facilities. They give it to the farm owners with license and not to the daily wage workers.”*
(G6, P3)

##### Alternate Employment

Some participants were willing to shift to alternate employment. However, there are limited employment opportunities in the village, so they are forced to work in tobacco farming. Some participants’ husbands do not allow them to work outside their village, so they are unable to take other work. For example, a participant noted, 


*“Even we feel like coming out of the village and working in garments, but the problem is our husbands will not allow us to do any other work outside the village.”*
(G5, P5)

However, participants were interested in rearing cows as mentioned by a participant, who said,


*“We can have a living if they get us a cow. We will give the milk to the dairy and get money.”*
(G3, P3)

Some participants also mentioned other limitations to shifting to alternate employment, such as not having a formal education and having limited employment opportunities in the village. For example, one participant said, 


*“If there is any company which can hire people like us who cannot read and write, they usually hire only literate people. If they provide an opportunity even, we can work in such companies.”*
(G5, P4)

##### Needs

Participants need monetary support during pregnancy, health care services from the tobacco board, and holidays. One participant mentioned, 


*“We are anemic after doing the tobacco work, so if they give us money, we will buy the medicines.”*
(G3, P6)

#### 3.3.4. Community

##### Health Care Services

Most participants used the government hospital for health care services, although a few participants used private health care services for C-section deliveries, other surgeries, and diagnostic procedures like ultrasound or x-rays. A participant said, 


*“Most of them go to big hospitals for Cesarean section. At that time they will need more money to pay the hospital bills, so if they give some money for the workers from the board it will be very useful for them.”*
(G1, P1)

Few participants reported home delivery for childbirth.

##### Health Care Providers

At the community level, participants received support from ASHAs, government hospital nurses, and doctors during their pregnancy and post-natal time period. Most of the occupational health information, such as the importance of wearing PPE, and reproductive health information, such as family planning, were provided by the health care providers. For example, one participant said, 


*“My husband would take me to the hospital. Sister (nurse) would give us all the information. She would tell us about the gap between children and operation (family planning). After delivery they came after seven months, and they advised for an operation. Since I have anemia, I got the operation late. She helped me in everything.”*
(G4, P2)

Another participant said, 


*“My mother-in-law and ASHA worker had helped me during pregnancy.”*
(G3, P4)

##### Needs

Participants needed transportation and ambulance services during medical emergencies. A participant mentioned, 

*“The transportation in our village is very poor. The ambulance people used to give us reasons when we called them. By the time they came to us, the delivery would have already happened.*”(G3, P4)

#### 3.3.5. Public Policy

At the societal level, tobacco farm laborer participants did not receive any support from the local, state, and national government and were in need of monetary and nutritional supplements.

##### Policies/Schemes

Participants stated that laws or policies were lacking as they did not receive any support for their health needs during pregnancy and the post-natal period working as women tobacco farm laborers.

##### Needs

From the local, state, and national governments, participants need monetary support during prenatal and post-natal periods and for sudden hospitalizations. Participants also need nutritional supplements for themselves and their families. They believe working in tobacco is making them anemic, so they are in need of nutritional supplements to manage it. One participant said, 


*“If we face any problem related to our health and we need to go the hospital and take the treatment there, but we do not have sufficient money, at that time the government should support for the women like us who depend on daily wages. I am suffering from stomach ache but I have never got it checked from the doctor as I am scared what if they ask me to undergo an operation. The government should help such people.”*
(G2, P7)

## 4. Discussion

### 4.1. Main Findings

This study explored the occupational and reproductive health experiences of women tobacco farm laborers during four distinct phases of their lifespan: menstruation, pregnancy, the post-natal period, and post-menopause. Our study participants worked in tobacco farming activities across their life course including during the reproductive phases of life. This study highlights participants’ needs for maternity and early childhood services, substance use counseling services, and monetary support. 

Tobacco farming involves intensive manual labor, and our participants reported musculoskeletal disorders in the lower back, wrists, shoulders, knees, and hips, which is similar to findings from a study on tobacco farmers in Thailand [46]. Chronic low back pain was a health problem among Brazilian tobacco farmers [47]. Tobacco farmers in Brazil were familiar with GTS, pesticide poisoning, and musculoskeletal disorders as health problems associated with working in tobacco farming [14]. In contrast, most participants in our study were not aware of GTS or these other issues. A relationship between work stress and menstrual disorders has been reported among tobacco farmers in Indonesia [15]. Our participants reported pain and discomfort during menstruation, However, we must further investigate the effects of nicotine and pesticide exposure on menstruation in this population.

A study conducted in rural China reported that maternal exposure to chemical fertilizers during pregnancy was associated with birth defects in infants [48]. In the study, participants mentioned being exposed to fertilizers during pregnancy, which, along with pesticide exposures, may explain the high incidence of adverse pregnancy and birth outcomes (Table 2). Pesticide-related symptoms such as loss of appetite, headache, eye irritation, dyspepsia/difficult digestion, skin allergy, and dizziness were reported by our participants while grading tobacco leaves, which could be due to exposure to pesticide residues on the tobacco leaves [22]. An exposure assessment could test this hypothesis and provide information needed to develop interventions.

Some participants believed tobacco use can cause harm, and most participants mentioned cancer as a health effect. They reported receiving this information from doctors in hospitals and seeing the anti-tobacco advertisements on tobacco packet covers, television, and radio. However, not all participants believed working in tobacco can cause harm. This could be due to limited health communication on the effects of occupational tobacco exposure. 

Organizational factors that influence tobacco farming include improved access to credit, well-developed supply chains, and governmental assistance [49,50]. The participants in these FGDs identified education, lack of employment opportunities in the villages, and spouses not allowing them to work in other villages as some of the barriers to shifting to an alternate employment. Providing adult education opportunities in the villages, as well as financial and technical support, such as accounting training, could increase the likelihood and viability of switching to alternative livelihoods [50].

### 4.2. Strengths and Limitations

The study findings were anchored to the widely adopted SEM to guide future research. However, this study has limitations. First, the FGDs were conducted in *Kannada* and translated to English, and some of the cultural context could have been lost in translation. Second, we lack the perspectives of stakeholders across other organizations and job roles; therefore, our study participants do not reflect a representative sample of tobacco farming in general.

### 4.3. Relevance of the Study and Policy Implications

Findings from this study will enable researchers to gain a deeper understanding of the impacts of tobacco farming on the livelihoods and reproductive and occupational health of the women farm laborers and their families. Our findings support the need for a holistic approach to improving the health of women tobacco farm laborers in India across individual, interpersonal, community, organizational, and broader societal levels. Our findings indicate the need for improved support for women tobacco farm laborers during menstruation, pregnancy, the post-natal period, and menopause, which are all critical reproductive phases of a woman’s life. Social policies and community interventions in these areas should be mindful of the needs of women tobacco farm laborers. Economically sustainable alternatives are needed to prevent possible adverse social and economic impacts on the women tobacco farm laborers, whose livelihoods depend on tobacco farming.

## 5. Conclusions

Women tobacco farm laborers reported GTS symptoms and musculoskeletal disorders during pregnancy and post-natal periods. Adverse pregnancy and birth outcomes like miscarriage, low birth weight, and preterm birth are reported among the participants. There is a need for monetary support during the critical reproductive phases of life, better knowledge of safe working practices, access to PPE among women, adult education, and alternate employment opportunities in this community. Policy makers should prioritize the needs of this under-served population while making tobacco policies and improving occupational health practices. 

## Figures and Tables

**Figure 1 ijerph-21-00606-f001:**
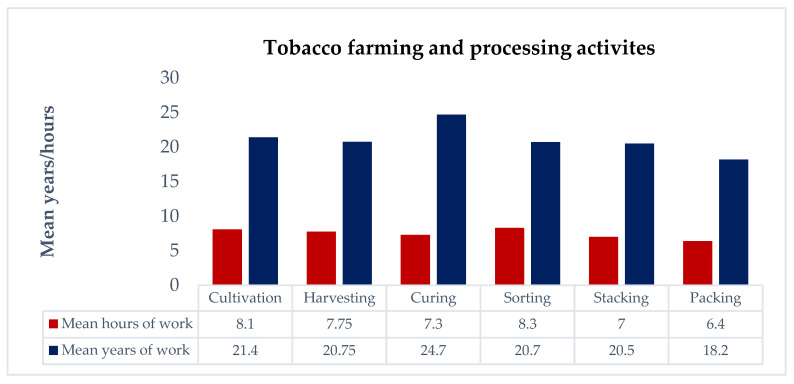
Duration of tobacco farming and processing activities.

**Figure 2 ijerph-21-00606-f002:**
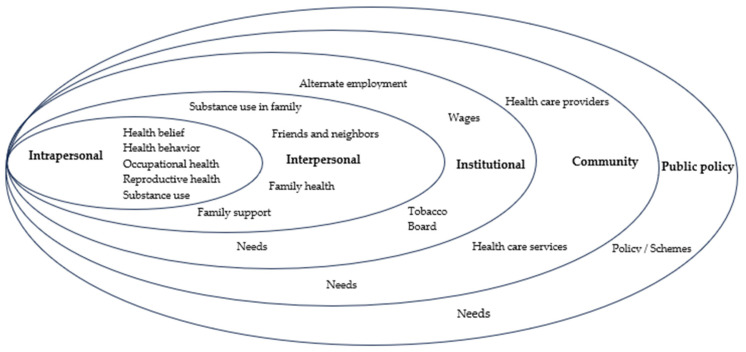
An overview of the themes that represent occupational and reproductive health challenges for women tobacco farmers across the five levels of the socioecological model.

**Table 1 ijerph-21-00606-t001:** Demographic characteristics of participants (*n* = 41).

Variable	*n* (%)
**Mean age (in years)**	38.22 ± 7.95
**Age groups**	
20 to 30 years	7 (17.1)
31 to 40 years	22 (53.7)
41 to 50 years	8 (19.51)
51 to 60 years	4 (9.76)
**Marital status**	
Married	33 (80.5)
Widowed	8 (19.5)
**Religion**	
Hindu	41 (100)
**Caste**	
Scheduled caste	30 (73.2)
Scheduled tribes	1 (2.4)
Other backward caste	3 (7.3)
General caste	7 (17.1)
**Education**	
Any college degree	1 (2.4)
Secondary school (class 11 and 12)	6 (14.6)
High school (class 6 to 10)	12 (29.3)
Primary school (class 1 to 5)	2 (4.9)
Literate but no formal education	1 (2.4)
Illiterate	19 (46.3)
**Annual Income (In Rupees)**	
Less than 20,000 (<240 USD *)	10 (24.4)
20,000 to 50,000 (240 to 600 USD *)	25 (61.0)
More than 50,000 (>600 USD *)	6 (14.6)
**Tobacco use**	
Smoking tobacco	0
Smokeless tobacco	6 (14.6)

* United States Dollar (USD).

**Table 2 ijerph-21-00606-t002:** Occupational and reproductive health of participants (*n* = 41).

Variable	*n* (%)
**Participants involved in t** **obacco farming activities**	
Cultivation	36 (87.8)
Harvesting	40 (97.6)
Curing	13 (31.7)
Sorting	36 (87.8)
Stacking	33 (80.5)
Packing	22 (53.6)
**Total number of pregnancies**	
≤2 pregnancies	30 (73.2)
>2 pregnancies	11 (26.8)
**History of abortion/miscarriage/death of child**	
At least one abortion	1 (2.4)
Miscarriage	5 (12.2)
History of death of child after birth to one year of age	4 (9.8)
**Tobacco exposure during pregnancy**	
Consumed tobacco during pregnancy	2 (4.9)
Tobacco farming during pregnancy	33 (80.5)
Exposed to tobacco dust during pregnancy	18 (43.9)
Wearing personal protection equipment (PPE) during work	6 (14.6)
**History of any allergy**	
During pregnancy	6 (14.6)
During post-natal period	5 (12.2)
In child 6 months after birth	8 (19.5)
In child 1 year after birth	9 (22.0)

## Data Availability

Data are not available due to the conditions of ethical approval. Participants were assured that raw data would not be shared.

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
