# Peer review of "Qualitative Study to Explore the Occupational and Reproductive Health Challenges among Women Tobacco Farm Laborers in Mysore District, India"

_ijerph, 2024, doi:10.3390/ijerph21050606_

Round 1

Reviewer 1 Report

Comments and Suggestions for Authors

The study seeks to examine the occupational challenges and reproductive health of women tobacco farm labors in a specific region of India. Overall, the study design and writing in various sections of the article are well executed. However, one critical aspect has been overlooked. Given that this study is qualitative in nature, ensuring the trustworthiness of the research findings is essential. This includes criteria such as credibility, dependability, transferability, and conformability, which were not addressed in the current manuscript. Therefore, it is imperative for the authors to provide a detailed explanation of how they ensured Trustworthiness in their research.

Author Response

Dear reviewer,

Thank you for your thoughtful review of our manuscript (ijerph-2971653), and for the helpful comments. I sincerely appreciate and thank you for taking your valuable time to help us substantially improve the paper. Below is the clarification for your comments.

Trustworthiness in this study is explained in terms of credibility, dependability, transferability, and conformability.

Credibility: Peer debriefing was done by the field researchers PR, SN, and MBN after every FGD with non-field coresearchers KM and PM with the notes taken during the discussions. These discussions helped us to identify the repeated themes and decide on the data saturation. During the study period, the field researchers visited the tobacco farms and tobacco barns to observe the tobacco farming and tobacco processing activities. Data triangulation in FGDs was done with five groups of tobacco farm laborers not owning tobacco fields and one group with tobacco farm owners who also work in the tobacco fields.

Transferability: While qualitative research tends to be more internally valid than generalizable in nature, we took time to compare our findings to the literature in the discussion section to demonstrate how our findings are in line with other study findings to attempt to demonstrate the transferability of our findings.

Dependability and confirmability: The audio recordings, notes, and transcripts are available for audits at PHRII. The data was coded by two authors (PR and MNB) and a third author (KM) resolved the conflicts. The study findings have been reviewed by senior authors PM, FvH, and LG.

(This is included in the main text and highlighted.)

Reviewer 2 Report

Comments and Suggestions for Authors

Dear Authors,

I read your paper with interest. 

I found two main issues in your research.

The number of subjects enrolled is very low and the study population is not representative.

Although, as you specify in the text, it is an exploratory study, so I hope you will overcome these issues in the future papers. 

I think it would also be useful to include controls (men and not pregnant women) to compare the results. 

Minor issue:

in the introduction, I suggest to move lines 59-64 before line 51 or 53. 

Best regards

Author Response

Dear reviewers,

Thank you for your thoughtful review of our manuscript (ijerph-2971653), and for the helpful comments of the three reviewers. I sincerely appreciate and thank you all for taking your valuable time to help us substantially improve the paper. 

Comments

Reply

Dear Authors,

I read your paper with interest.

I found two main issues in your research.

The number of subjects enrolled is very low and the study population is not representative. Although, as you specify in the text, it is an exploratory study, so I hope you will overcome these issues in the future papers.

Thank you for your comments.

This is a qualitative exploratory study; we stopped collecting the data once there was data saturation. We will include more participants in future studies.

I think it would also be useful to include controls (men and not pregnant women) to compare the results.

Thank you for the suggestions.

We will include men in our future studies. Our study participants are non-pregnant women.

Minor issue:

in the introduction, I suggest to move lines 59-64 before line 51 or 53.

Thank you for the suggestion. We have changed the order of the lines and it is highlighted in the main document.

Reviewer 3 Report

Comments and Suggestions for Authors

1. In line 27 of the abstract, it is mentioned that the majority of Indian tobacco farm laborers are women and children. However, upon reading the background section, it is noticed that the impact of GTS on child labor is not discussed at all. Therefore, it would be crucial to provide more background information to strengthen the issue and ensure that this study is suitable for publication.

2. Could you provide information on the potential impact of workplace conditions on reproductive health? Specifically, are there any dangers faced by female workers and children that extend beyond menstrual disorders? Are there other reproductive health issues that could pose a significant risk to workers? (Introduction section)

3. What harmful substances are in tobacco chemical fertilizers and pesticides? How are workers exposed? (Line 84)

4. Is there data available on occupational and reproductive health issues among female workers or children in the study location? If so, please include it in the introduction section.

5. In the introduction section, it would be helpful to explain what measures tobacco farmers, companies, or local governments have taken to minimize the impact on reproductive health. For instance, the use of Personal Protective Equipment (PPE), environmental assessments to detect hazardous material contamination or any other relevant controls.

6. In line 114 the author mentions a random selection of participants in 6 villages. Can you explain in more detail how to carry out randomization?

7. Why were pregnant women in this study excluded from study participants? Can you explain in more detail?

8. In the abstract, it is stated that child workers are one of the vulnerable groups involved in the work of tobacco farmers. Can you explain why child workers were not participants in this study?

9. In lines 225-243, the text simply repeats what is written in Table 2. Please summarize only the important findings from Table 2

10. Please review the conclusion to ensure alignment with the problem statement and study findings, particularly in the area of reproductive health.

Author Response

Dear reviewer,

Thank you for your thoughtful review of our manuscript (ijerph-2971653), and for the helpful comments of the three reviewers. I sincerely appreciate and thank you all for taking your valuable time to help us substantially improve the paper. We have made substantial edits to the entire manuscript, as shown in the revised version with changes highlighted. Below please find our responses to the reviewers' specific comments.

Comments

Reply

1. In line 27 of the abstract, it is mentioned that the majority of Indian tobacco farm laborers are women and children. However, upon reading the background section, it is noticed that the impact of GTS on child labor is not discussed at all. Therefore, it would be crucial to provide more background information to strengthen the issue and ensure that this study is suitable for publication.

We have cited two articles that reported the GTS symptoms among children, and it is highlighted in the main document.

The GTS symptoms have been reported among children 12 to 17 years of age in Kentucky Regional Poison Center [7]. A case control study conducted in 1993, reports children younger than 16 years of age represent 9% GTS reported [8].

7. McKnight, R.H.; Spiller, H.A. Green Tobacco Sickness in Children and Adolescents. Public Health Rep. Wash. DC 1974 2005, 120, 602–605, doi:10.1177/003335490512000607.

8. Ballard, T.; Ehlers, J.; Freund, E.; Auslander, M.; Brandt, V.; Halperin, W. Green Tobacco Sickness: Occupational Nicotine Poisoning in Tobacco Workers. Arch. Environ. Health 1995, 50, 384–389, doi:10.1080/00039896.1995.9935972.

2. Could you provide information on the potential impact of workplace conditions on reproductive health? Specifically, are there any dangers faced by female workers and children that extend beyond menstrual disorders? Are there other reproductive health issues that could pose a significant risk to workers? (Introduction section)

 Almost 5% of failure in hearing screening was found among the infants whose mothers worked with pesticides in tobacco fields [16]. Tobacco farmers exposed to pesticides exhibited signs of central auditory dysfunction characterised by decrements in temporal processing and binaural integration processes [17].

16. Buaski, J.P.; Magni, C.; Fujinaga, C.I.; Gorski, L.P.; De Conto, J. Exposure of Tobacco Farm Working Mothers to Pesticides and the Effects on the Infants’ Auditory Health. Rev. CEFAC 2018, 20, 432–441, doi:10.1590/1982-021620182042218.

17. França, D.M.V.R.; Bender Moreira Lacerda, A.; Lobato, D.; Ribas, A.; Ziliotto Dias, K.; Leroux, T.; Fuente, A. Adverse Effects of Pesticides on Central Auditory Functions in Tobacco Growers. Int. J. Audiol. 2017, 56, 233–241, doi:10.1080/14992027.2016.1255787.

3. What harmful substances are in tobacco chemical fertilizers and pesticides? How are workers exposed? (Line 84)

Pesticides are essential for tobacco growing, as “the crop could not be produced economically without them” according to tobacco industry documents [18]. Farmers are exposed to pesticide residues through ingestion, inhalation, and skin contact with dry tobacco leaves during work [19]. Organochlorine and organophosphate pesticides are widely used in tobacco farming [20]. Dichlorodiphenyltrichloroethane (DDT) is an organochlorine pesticide known for its long-term persistence in the environment and its irreversible toxic health effects causing neurological damage, endocrine disorders, and reproductive health effects [20,21].

Some pesticides are endocrine disrupting chemicals (EDCs) which mimic hormones re-sulting in hormonal disruption [24].

4. Is there data available on occupational and reproductive health issues among female workers or children in the study location? If so, please include it in the introduction section.

We have added relevant information found in the literature to address these points, see lines 137-150 in the introduction section.

5. In the introduction section, it would be helpful to explain what measures tobacco farmers, companies, or local governments have taken to minimize the impact on reproductive health.

For instance, the use of Personal Protective Equipment (PPE), environmental assessments to detect hazardous material contamination or any other relevant controls.

We have added relevant information found in the literature to address these points, see lines 151-157 in the introduction section.

6. In line 114 the author mentions a random selection of participants in 6 villages. Can you explain in more detail how to carry out randomization?

The six villages (study location) were randomly selected. The participants were recruited through convenience sampling with the help of the ASHA workers and the staff from PHRII. (Line 147)

7. Why were pregnant women in this study excluded from study participants? Can you explain in more detail?

Pregnant women were not included in this study because they are vulnerable population.

8. In the abstract, it is stated that child workers are one of the vulnerable groups involved in the work of tobacco farmers. Can you explain why child workers were not participants in this study?

The goal of this study is to focus on the reproductive and occupational health of the women tobacco farmers. We wanted to highlight along with women, their children are also involved in tobacco farming and are exposed to tobacco in-utero during pregnancy. We will plan to include children in our future studies.

9. In lines 225-243, the text simply repeats what is written in Table 2. Please summarize only the important findings from Table 2

We have shortened this section to focus only on important findings.

10. Please review the conclusion to ensure alignment with the problem statement and study findings, particularly in the area of reproductive health.

Thank you. Included some salient findings.

Adverse pregnancy and birth outcomes like miscarriage, low birth weight, and pre-term birth are reported among the participants. There is a need for monetary support during the critical reproductive phases of life, need to improve the knowledge of safe working practices, access to PPEs among women, adult education, and alternate employment opportunity in this community.

Once again, we thank you and the reviewers for your thoughtful review process and for helping us to significantly improve the manuscript.

                                                                        Sincerely,

                                                                        Priyanka Ravi
